# Prevalence, Risk Factors and Vaccine Response against Hepatitis B in People Aged 50 Years or Older

**DOI:** 10.3390/vaccines11030597

**Published:** 2023-03-05

**Authors:** Luana Rocha da Cunha Rosa, Leyla Gabriela Verner Amaral Brandão, Winny Éveny Alves Moura, Lays Rosa Campos, Grécia Carolina Pessoni, Juliana de Oliveira Roque e Lima, José Cássio de Moraes, Megmar Aparecida dos Santos Carneiro, Sheila Araújo Teles, Karlla Antonieta Amorim Caetano

**Affiliations:** 1Faculty of Nursing, Federal University of Goias, Goiânia 74605-080, GO, Brazil; 2Municipal Health Secretariat, Goiânia 74884-900, GO, Brazil; 3Faculty of Medical Sciences of Santa Casa de São Paulo, São Paulo 01224-001, SP, Brazil; 4Institute of Tropical Pathology and Public Health, Federal University of Goias, Goiânia 74690-900, GO, Brazil

**Keywords:** hepatitis B vaccines, middle aged, elderly, epidemiology, immunogenicity vaccine

## Abstract

Universal immunization against hepatitis B has contributed to reducing incidence of the disease, but older individuals remain susceptible to acquiring the hepatitis B virus worldwide. Thus, this study aimed to investigate the epidemiology of HBV infection in individuals aged 50 years and over in central Brazil and to evaluate the immunogenicity of the monovalent vaccine against hepatitis B in this age group using two vaccine regimens. Method: Initially, a cross-sectional and analytical study was carried out to investigate the epidemiology of hepatitis B. Then, individuals without proof of vaccination for hepatitis B were recruited for a phase IV randomized and controlled clinical trial using two vaccine regimens: Intervention Regimen (IR) (three doses of 40 μg at months 0, 1 and 6) vs. Comparison Regimen (CR) (three doses of 20 μg at months 0, 1 and 6). Results: The overall prevalence of exposure to HBV was 16.6% (95% CI: 14.0%–9.5%). In the clinical trial, statistical differences in protective titers were observed (*p* = 0.007; IR 96% vs. CR 86%) and the geometric mean of anti-HBs titers was higher in individuals who received the IR (518.2 mIU/mL vs. 260.2 mIU/mL). In addition, the proportion of high responders was higher among those who received the IR (65.3%). Conclusion: reinforced doses should be used in individuals aged 50 years or older to overcome the lower efficacy of the vaccine against hepatitis B.

## 1. Introduction

Hepatitis B (HBV) infection threatens public health globally [1]. It is estimated that, in 2019, there were 316 million (95% CI: 284%–351%) chronic disease carriers in the world at risk of developing complications such as cirrhosis and hepatocellular carcinoma. In that same year, there were approximately 555,000 (95% CI: 487,000–630,000) global deaths related to HBV, representing 48.8% (95% CI: 44.6%–52.7%) of all deaths from hepatitis [2].

The development and availability of the first hepatitis B vaccine in the 1980s [3] was a milestone for prevention and currently remains the gold standard strategy for eliminating this infection [4]. In several countries, the HBV vaccine’s introduction occurred gradually according to age groups and risk groups and, in general, starting with newborns and children [5,6]. It is estimated that in 2019, 85% of children were vaccinated with the complete vaccine schedule worldwide [7].

Universal immunization, focusing mainly on children and adolescents, has contributed to reducing the incidence and changing the epidemiological profile of HBV [1], leading to a shift in the epidemiological curve of the infection towards more vulnerable older individuals at the beginning of the 21st century [8,9]. On the other hand, the increase in cases of hepatitis B in the elderly is multifactorial and unsafe sexual practices are associated with the acquisition and dissemination of Sexually Transmitted Infections (STIs) [10,11]. It is believed that the better sexual performance provided by drugs for the treatment of erectile dysfunction, combined with older adults being socially represented as asexual, has contributed to unprotected sexual intercourse, multiple sexual partners, sexual intercourse with sex workers, and other risk factors for STIs [10,11].

Currently, Brazil has low endemicity for hepatitis B. However, studies indicate higher rates of HBV exposure in older adults than in children and adolescents [8]. The hepatitis B vaccine was included in the Brazilian National Immunization Program in the late 1980s, initially for children and gradually expanded to adolescents and adults. However, the vaccine was made available free of charge to the entire population, regardless of age, only in 2015 [12]. Therefore, vaccination coverage in older adults is still incipient [13,14] and, associated with unsatisfactory vaccine performance in more advanced age groups, represents a challenge for public health.

Studies conducted with middle-aged and elderly adults showed lower vaccine response against hepatitis B using the conventional scheme [15,16]. On the other hand, the reinforced regimen is reported in the literature as a successful strategy to induce the production of anti-HBs protective titers in individuals with a lower vaccine response [17,18].

The recombinant hepatitis B vaccine is produced with different expression systems to obtain HBsAg. These vaccines have good immunogenicity compared with the gold standard (Engerix-B or Recombivax-HB vaccine) [19,20]. Brazil has used vaccines that use Hansenula polymorpha as an HBsAg expression system in its National Immunization Program. However, data on vaccine response against hepatitis B in elderly populations are scarce and, until the cohort of children immunized at birth reaches late adulthood, strategies that overcome this limitation are necessary. Thus, reinforced doses of recombinant hepatitis B vaccine may be a safe alternative for the elderly.

Therefore, the proposal is to investigate the epidemiological profile of HBV and evaluate the immunogenicity of the hepatitis B vaccine in adults aged 50 years or older, using conventional doses (20 μg) vs. boosted doses (40 μg). We believe that the results of this study can support health managers in decision making about more effective hepatitis B vaccine regimens for the elderly.

## 2. Materials and Methods

The present investigation was conducted in two stages. Initially, a cross-sectional study investigated the epidemiology of hepatitis B in adults aged 50 years or older. Then, individuals without vaccination evidence were invited to participate in a phase IV randomized and controlled clinical trial to compare the response with the monovalent vaccine against hepatitis B, using a reinforced regimen vs. the conventional scheme.

The study participants were adults aged ≥50, residing in a small municipality and the capital of the State of Goiás, located in midwestern Brazil. The small municipality, Goiandira, is located in the southwest of Goiás at a distance of 266 km from the capital. In 2010, in the last census, the estimated population was 5265 inhabitants, of which 27.8% were aged 50 or over. Goiandira is ranked fourth in Goiás in the Human Development Index (HDI), with a value of 0.760 [21].

Goiânia, the state capital, is ranked highest in the state concerning the HDI at 0.799. It has 1,302,001 inhabitants identified in the last census (2010) and about 1,297,076 live in the urban area. Of the total, 19.5% were 50 years old or older [21].

### 2.1. Cross-Sectional Study

For the cross-sectional study, the minimum sample required would be 580 participants, considering a statistical power of 80% (β = 20%), a significance level of 95% (α < 0.05), a precision of 5%, a design effect of 2.5 and a prevalence of exposure to HBV in individuals aged 40 years or older residing in the midwest region of Brazil of 18.5% [22]. Ultimately, 682 individuals were part of the cross-sectional study.

Inclusion criteria were: being a resident of the urban or rural area of the municipality of Goiandira-GO or residing in the urban area of Goiânia-GO, in addition to being 50 years old or older. In addition, individuals who were under the influence of psychoactive drugs at the time of the interview or data collection or had a record of a diagnosis of Alzheimer’s disease or other types of dementia and cognitive alteration through the manifestation of forgetfulness, loss of attention and reduced ability to solve problems reported by family members/health team or perceived by the interviewer during the presentation of the research objectives were excluded.

The convenience sampling method was used and data collection took place from July 2017 to December 2019. Initially, partnerships were consolidated between the team of researchers and members of the health departments of the municipalities involved. Based on this prior planning, the dates for data collection and the means of disseminating the research were established, which included the distribution of explanatory pamphlets about the importance of participating in the project and in-person invitations through the community health agents to individuals who met the criteria of eligibility.

In both municipalities, those who attended on the established dates at the defined collection locations (dependencies of health units, churches or community centers) were invited to participate in the study.

Then, after confirmation of the participant’s eligibility, authorization to participate in the study was requested voluntarily through the written signature of the Informed Consent Form (ICF). In cases of illiterate individuals, the ICF was read to the participants and a witness and the participant’s signature was fingerprinted. 

All were instructed about the importance, objectives, risks and benefits of participating in this study and the freedom to leave at any time. Adherence to the study was high, approximately 100%. Only two individuals refused to participate.

First, participants were invited to an individual and private interview using an adapted semi-structured script containing questions about sociodemographic aspects, disease history, current health conditions, use of licit and illicit drugs, sexual and parenteral risk behaviors, violence, discrimination and vaccination status. After this step, 10 mL of blood was collected by peripheral venipuncture.

For hepatitis B screening, 1 mL of blood from participants was used at the time of data collection. For this step, a marker test (rapid test) was used to detect the HBsAg marker (Quibasa-Bioclin, Belo Horizonte, Brazil).

The rest of the blood obtained was identified with the number of participants, placed in climate-controlled thermal boxes and transported to laboratories. In Goiânia, blood samples were centrifuged and the serum obtained was stored at −20 °C at the Multi-User Clinical Research Laboratory (LAMPEC) of the Faculty of Nursing/UFG. In Goiandira, blood samples were processed at the municipal laboratory and the serum was stored in a −20 °C freezer, then transported to LAMPEC in Goiânia. At this stage, the integrity of the samples was guaranteed, as well as the biosafety norms for transporting samples, as recommended by the Brazilian National Health Surveillance Agency. Thus, all serum samples were stored in a −20 °C freezer in LAMPEC until the tests were performed.

The serum were tested for hepatitis B virus serological markers: anti-HBc, anti-HBs, and HBsAg. The Enzyme-Linked Immunosorbent Assay (ELISA (Quibasa-Bioclin, Belo Horizonte, Brazil)) was used to detect the HBsAg serological marker. In addition, the Chemiluminescent Microparticle Immunoassay (CMIA (Abbott Laboratories, Rio de Janeiro, Brazil)) was used for the anti-HBc to anti-HBs marker. The exams were carried out at LAMPEC and in partner private laboratories.

The outcome variables of the cross-sectional study were:Exposure to HBV: positivity for markers of exposure or infection by HBV: HBsAg or anti-HBc;Serological profile of vaccination: isolated positivity for the anti-HBs marker;Reported on the vaccination card: three doses of vaccine against hepatitis B on the vaccination card;Serological profile of vaccination or vaccine reported on the vaccination card: positivity for the isolated anti-HBs marker or three vaccine doses against hepatitis B via vaccination card.

Interview data and test results were analyzed using the STATA statistical package version 13.0 (StataCorp, College Station, TX, USA). Descriptive analysis was performed using frequency distributions, means and standard deviations. In addition, Chi-square and Fisher’s exact tests were used to test differences between proportions. Finally, prevalences were calculated with a 95% confidence interval.

Multiple analysis was performed for the outcome variable “exposure to HBV”. The investigated independent variables that presented *p* < 0.250 in the bivariate analysis were submitted to multiple analysis by logistic regression, using the forward method as a selection criterion. *p* values < 0.05 were considered significant. The model’s goodness of fit was performed using the Hosmer–Lemershow test and the ROC curve.

### 2.2. Randomized and Controlled Clinical Trial

The clinical trial was nested within the cross-sectional study. For logistical reasons and to optimize the individual’s participation in the study, the inclusion criteria were to receive the first dose of the vaccine: Having participated in the cross-sectional study;Not having a vaccination record or report of previous vaccination against hepatitis B;Having presented a negative result for HBsAg in the rapid test performed in the cross-sectional study.

Individuals who were positive for hepatitis B markers (anti-HBs or anti-HBc) in the serological screening of the cross-sectional study were excluded and did not receive subsequent doses. The participant’s vaccination status was defined as having a serological profile of vaccination, three doses of the vaccine against hepatitis B on the vaccine card and both (serological profile of vaccination or as reported on the vaccination card).

For the clinical trial, it was considered that 60% [15] of individuals aged 50 years or older would respond to the conventional vaccine and 80% to the reinforced dose, with a statistical power of 80% (β = 20%) and a significance level of 95% (α < 0.05), and 30% were added to the calculation to replace losses. Therefore, 240 people would be needed: 120 for the intervention group and 120 for the comparison group.

Recruitment started with the execution of the cross-sectional study, as shown in Figure 1. Among the 682 participants, 434 middle-aged and elderly adults were enrolled in the clinical trial. In the end, the sample consisted of 238 individuals: 124 from the intervention group and 114 from the control group.

For the allocation sequence to achieve a 1:1 ratio, random numbers were generated in blocks (every 20 individuals) through the website https://www.random.org/ (accessed on 15 June 2017), placed in opaque envelopes and sealed and numbered by a member of the team who did not participate in the data collection stage.

The IR group received 3 doses of 40 μg (2 mL) of hepatitis B vaccine at months 0, 1 and 6, while the CR group received 3 doses of 20 μg (1 mL) of the vaccine in months 0, 1 and 6. About 30 to 60 days after the last dose of the vaccine, 5 mL of blood was collected from the participants to evaluate the vaccine response against hepatitis B. For quantitative detection of anti-HBs, the CMIA was used.

The “Hepatitis B vaccine (rDNA)” vaccine, produced in Hansenula polymorpha yeast cells from the Serum Institute of India laboratory (Pune, India), was used. The vaccines were provided by the Health Department of the State of Goiás and the lots used were: 035L6030, 035L7007A and 035L7012.

Anti-HBs titers ≥10 mIU/mL were considered protective. Thus, vaccinees were classified as nonresponders (anti-HBs <10 mIU/mL), low (anti-HBs: 10–100 mIU/mL), good (anti-HBs: 100–999 mIU/mL) and high (anti-HBs ≥ 1000 mIU/mL) responders [23,24].

The primary outcome of the study was: the success of the proposed procedure, defined by developing isolated anti-HBs titers (≥10 mIU/mL) after three boosted doses (40 μg) of the hepatitis B vaccine, evaluated in the period from 30 to 60 days after completing the vaccination regimen, in a significantly higher percentage than the comparison regimen.

Descriptive analysis was performed using frequency distribution, arithmetic means and standard deviations. The geometric mean of anti-HBs titers (GMT) for seroprotection and seroconversion for the hepatitis B vaccine was calculated with a 95% confidence interval. Student’s *t*-test was used to compare means. In addition, the Chi-square test and Fisher’s exact test were used to test the significance of differences between proportions and to assess the relationship between seroprotection and the type of vaccine regimen received. *p* values < 0.05 were considered statistically significant.

## 3. Results

### 3.1. Cross-Sectional Study

Of the 682 participants investigated, 113 had at least one serological marker of exposure to HBV, resulting in an overall prevalence of 16.6% (95% CI: 14.0%–19.5%). In addition, the HBsAg marker was identified in three individuals (0.4%; 95% CI: 0.1%–1.3%) and the marker referring to previous vaccination against hepatitis B (isolated anti-HBs) in 99 (14.5%; 95% CI: 12.1%–17.4%). Of the total, 470 (68.9%; 65.3%–72.3%) individuals aged 50 years or older were susceptible to HBV infection.

Table 1 presents the bivariate and multiple analysis of sociodemographic characteristics, lifestyle habits and sexual and parenteral risk behaviors associated with exposure to HBV (serological positivity for any exposure marker). The variable “history of sexual intercourse with a sex worker” (OR = 1.9; 95% CI: 1.1%–3.3%) was a predictor of exposure to HBV (*p* < 0.05). In addition, there was a marginal significance between “sexual intercourse with a person of the same sex” (OR = 3.1; 95% CI: 0.96%−10.3%) and the outcome.

The rates for the serological profile of vaccination, reporting on the vaccination card (three doses) and serological profile of vaccination or reporting on the vaccination card were 15% (95% CI: 12.1%–17.4%), 21, 8% (95% CI: 18.9%–25.1%) and 29.6% (95% CI: 26.3%–33.2%), respectively. Considering the individuals who presented three doses of the vaccine against hepatitis B through the vaccination card (n = 147), the majority (69.1%; n = 103) did not present isolated anti-HBs protective titers. Table 2 presents the characteristics of the individuals who reported the three doses of the hepatitis B vaccine, according to the vaccine response. For this analysis, two participants were excluded because they had the date of the last vaccine dose <60 days from the date of collection. It can be observed that there was a statistical difference between the groups (*p* < 0.05) considering the variables: “positive report of systemic arterial hypertension” (*p* = 0.002) and “age ≥ 60 years” (*p* = 0.007).

### 3.2. Randomized and Controlled Clinical Trial

Table 3 shows that the intervention and comparison groups are comparable, considering sociodemographic and clinical characteristics (*p* > 0.05).

The response to the hepatitis B vaccine according to the Intervention Regimen and the Comparison Regimen is presented in Table 4.

Scheme 96. 95% CI: 90.9–98.3) vs. CR (86%; 95% CI: 78.4–91.2) (*p* = 0.007). Therefore, the geometric mean of titers was higher in individuals who received IR (518.2 mIU/mL; 95% CI: 456.0 mIU/mL–580.4 mIU/mL) vs. CR (260.2 mIU/mL; 95% CI: 183.5 mIU/mL–336.9 mIU/mL). In addition, the proportion of high responders was higher among those who received the Intervention Regimen (65.3%; 95% CI: 56.6–73.1) than among participants who received the concurrent Comparison Regimen (46.5%; 95% CI: 37.6–55.6). It is important to mention that no side effects were observed during the interventional study.

## 4. Discussion

In this investigation, the HBV exposure rate was 16.6% (95% CI: 14.0%–19.5%), higher than the prevalence identified in the young adult Brazilian population, aged between 20 and 39 years (6.6%; 95% CI: 6.0%–7.3%), and similar to that identified in middle-aged and elderly individuals, aged 40 years or older (16.5%; 95% CI: 15.4%–17.6%), according to a population-based study conducted in all regions of Brazil [22].

On the other hand, high rates of exposure to HBV in older individuals can be observed in different regions of Brazil. For example, in the midwest, an investigation carried out in a district in the interior of Mato Grosso estimated the prevalence of anti-HBc at 46.7% (95% CI: 32.8%–55.2%) in 71 individuals aged equal to or greater than 50 years [25]. In the north region, in Tocantins, a study carried out with individuals residing in a small municipality and with Amerindian tribes in the eastern Amazon determined a rate of exposure to HBV of 34.1% (95% CI: 25.2%–44.3%) in 91 older adults (>60 years) [26]. Similarly, in the western Amazon, an investigation carried out in the interior of the state of Acre estimated the prevalence of anti-HBc at 49.1% (95% CI: 41.6%–56.7%) in 165 adults aged ≥50 years [27].

In other countries with low endemicity, high exposure to HBV in the elderly can also be observed. For example, a retrospective study carried out in a university hospital in southern Italy showed a rate of anti-HBc of 44.8% (95% CI: 41.0%–48.6%) in 652 individuals aged 61 years or older [28]. In India, a population-based survey detected an anti-HBc exposure rate of 32% (95% CI: 28.6%–35.6%) in 823 people aged 60 years or older [29].

Contrary to what is pointed out in other investigations that present factors associated with HBV related to social characteristics, parenteral and vertical risk behaviors [30,31], in our study, only sexual risk characteristics were predictors of exposure to the virus. We believe that this result confirms what was expected for a region of low endemicity and a group with characteristics of the general population of Brazil.

It is also important to mention that some behaviors and social characteristics also had epidemiological importance among those exposed to HBV. Among individuals who reported less than five years of study, 19.0% were positive for exposure to HBV. On the other hand, considering the parenteral risk variables, we can highlight that history of incarceration, history of hemodialysis and shared sharp personal hygiene objects also showed relevant exposure rates. These variables are found in other studies and have historically been risk factors for exposure to the hepatitis B virus [30,31,32].

Therefore, in the present study, the variable “history of sexual intercourse with a sex worker” was associated with exposure to HBV (adjusted OR: 1.9; *p* = 0.013). In addition, hiring sexual services raises the risk of acquiring STIs [33,34]. This investigation confirms that individuals who reported a history of sexual intercourse with a sex worker were 1.9 times more likely to be exposed to HBV. On the other hand, commercial sex has been associated with irregular condom use, multiple sexual partners, sexual contact under the influence of alcohol and drugs and a low perceived risk of acquiring STIs [33,34]. Because of this, sex workers are considered a key population for the spread of hepatitis B [33,35].

“History of a same-sex relationship” was marginally associated with exposure to HBV (adjusted OR: 3.1; *p* = 0.059), suggesting that the sample size was insufficient to show a statistical difference. However, several studies in the world and in Brazil have identified a high frequency in these individuals of sexual risk behaviors closely related to the dissemination of HBV and other STIs, emphasizing unprotected receptive anal sex [36,37,38].

Three individuals in our study were positive for the serological marker HBsAg, indicating the presence of infection. All lived in the small town and were aged 60 or over. None reported having a steady sexual partner, but one participant confirmed having had sex in the last 12 months without using a condom. It highlights sexual risk behaviors in maintaining the hepatitis B transmission chain, contributing to the epidemiology of the internalization of STIs in our country. All these participants were referred for confirmation of the diagnosis and, if necessary, treatment and follow-up.

This study used records (the vaccination card) and the serological profile of immunization to estimate vaccination coverage in older adult individuals. Although it was observed that only 15% were considered immunized using the serological profile information, this percentage increased to 21.8% when the vaccination card was considered, reaching 29.6% when the two sources were considered.

Unfortunately, these data are similar to other national studies on the elderly [26,39]. These low vaccination rates against hepatitis B demonstrate the need to expand efforts to immunize the adult population in the middle and late stages. An effective strategy to expand access and to improve the hepatitis B vaccination situation suggests taking advantage of vaccination opportunities that already mobilize many older adults, such as the national annual flu shot campaigns.

In Brazil, other key places for vaccination against hepatitis B are the reference centers for health care for the elderly. These health units promote healthy aging, maintain and rehabilitate functional capacity and provide other health needs for elderly patients [40]. Therefore, offering the hepatitis B vaccine in these environments with more older adults can expand vaccination coverage in the specific age group and increase vaccine follow-up efficiency.

Among the 149 demonstrably vaccinated individuals in the present study, the majority (69.1%; n = 103) did not show protective titers of isolated anti-HBs. Based on these data, it is not possible to state whether these individuals did not respond to primary vaccination or whether they lost protective anti-HBs titers over time. However, the mean time, in years, between the date of the last dose of the vaccine and the date of the test for the anti-HBs marker in our study was approximately five years, reasonably enough time to be able to identify protective titers, since research shows the persistence of vaccine anti-HBs for approximately 30 years in healthy younger individuals [41,42]. This result reinforces the need for follow-up and monitoring of the vaccination schedule in the elderly, checking for vaccine response after a complete schedule by performing the anti-HBs test.

Furthermore, in this same vaccinated group, who presented a vaccination card with three doses of the hepatitis B vaccine (n = 149), it was possible to identify variables related to non-vaccination response or possible loss of anti-HBs titers after the vaccination. Reports of systemic arterial hypertension and older age were characteristics significantly associated with the absence of protective bonds.

Systemic arterial hypertension causes mechanical and oxidative damage to blood vessels and target organs, stimulating mechanisms that activate the innate and adaptive immune system. These alterations produce chronic inflammatory processes that can influence vaccine responses [43,44].

The lower efficacy of hepatitis B vaccines in older individuals has been described in the literature and is a worrying phenomenon. Studies conducted worldwide in the age group of 50 years or older have shown higher rates of non-response or lower vaccine responsiveness after a complete primary immunization schedule [15,16]. In addition, aging of the immune system negatively interferes with the vaccine response against hepatitis B, causing several changes in the individual’s immune response [16,45].

Because of this, the present study shows an effective strategy to increase the rate of older adults protected against hepatitis B in an unprecedented way. In this investigation, individuals aged 50 years or more who received a double dose of the vaccine against hepatitis B presented a significantly higher geometric mean of anti-HBs titers (*p* = 0.007) compared with the comparison group (IR = 518.2 (95% CI: 456.0%–580.4%) vs. CR = 260.2 (95% CI: 183.5%–336.9%)). In addition, participants immunized with the intervention regimen had higher proportions of high responders (IR: 65.3% vs. CR: 46.5%).

It is interesting to discuss that the expression systems used in manufacturing vaccines can influence the production of protective titers against hepatitis B [46,47]. Considering the two central expression systems used in second-generation vaccines, Hansenula polymorpha and Saccharomyces cerevisiae, available data are similar for both vaccines and show low immune response in older groups [16,48]. Thus, in the absence of studies investigating the efficacy of double doses of hepatitis vaccines, especially in the elderly, the results presented here may contribute to changing public policies in all countries, regardless of which vaccines are on the schedule.

This study has some limitations. As this is an older population, intimate/sexual issues could cause some embarrassment and interfere with the authenticity of the information. On the other hand, the entire research team was instructed to conduct the interviews privately and ethically. In addition, at the beginning of the interview, the participants were duly informed about the confidentiality of the responses. Memory biases may also have occurred. Alternatively, the interviewers were trained to contribute by stimulating the participants’ memory. As for the clinical trial, although the study responded to the proposed objective (evaluating the effectiveness of vaccine regimens), other variables of great epidemiological importance, such as the analysis of the kinetics of anti-HBs titers during sample follow-up and the factors of risk of non-responders to the vaccine, were not investigated due to the limitation of the final sample.

## Figures and Tables

**Figure 1 vaccines-11-00597-f001:**
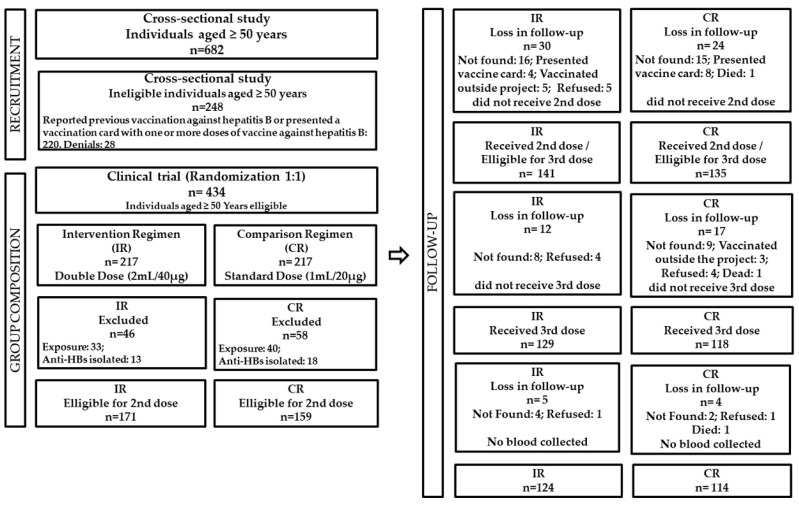
Flowchart of the randomized and controlled clinical trial carried out with 434 individuals who received the vaccine against hepatitis B according to an Intervention Regimen (IR) or Comparison Regimen (CR), Goiás, 2017−2020.

**Table 1 vaccines-11-00597-t001:** Bivariate and multiple analysis of sociodemographic characteristics, lifestyle habits and risk behaviors associated with exposure to HBV in 682 individuals aged ≥ 50 years living in a small municipality and capital of the State of Goiás, 2017−2019.

Variable	Bivariate Analysis						Multiple Analysis ****	
							HBV (n = 682)	
	Total	Positive	Negative	*p*	OR * (CI 95% **)		*p*	OR * (CI 95% **)
	n= 682 (%)	n =113 (%)	n = 569 (%)					
Sex				0.094			0.094	
Male	214 (31.4)	43 (20.0)	171 (80.0)		1			
Female	468 (68.6)	70 (15.0)	398 (85.0)		0.69 (0.46−1.06)			
Age				0.135				
50–59 years old	266 (39.0)	37 (86.1)	229 (13.9)		1			
≥60 years	416 (61.0)	76 (18.3)	340 (81.7)		1.38 (0.90−2.12)			
City of Origin				0.514				
Capital	339 (49.7)	53 (15.6)	286 (84.4)		1			
Interior	343 (50.3)	60 (17.5)	283 (82.5)		1.14 (0.76−1.71)			
Education/(NI = 5) ***			0.168					
> 9 years	184 (27.2)	24 (12.5)	161 (87.5)		1			
5–9 years	183 (27.0)	30 (18.4)	153 (81.6)		1.37 (0.76−2.47)			
<5 years	310 (45.8)	59 (19.0)	251 (81.0)		1.65 (0.98−2.77)			
Internet				0.104				
No	369 (54.1)	69 (18.7)	300 (81.3)		1			
Yes	313 (45.9)	44 (14.1)	269 (85.9)		0.71 (0.47−1.1)			
Religion/(NI = 2) ***			0.092					
Without religion	33 (4.9)	9 (27.3)	24 (72.7)		1			
Religious	647 (95.1)	104 (16.1)	543 (83.9)		0.51 (0.23−1.13)			
Self-declared color				0.795				
White	236 (34.6)	39 (16.5)	197 (83.5)		1			
Brown and Black	135 (63.8)	73 (16.8)	362 (83.2)		1.02 (0.67−1.56)			
Yellow (Eastern) and Red (Indigenous)	11 (1.6)	1 (9.1)	10 (90.9)		0.51 (0.63−4.06)			
Marital Status/(NI = 1) ***			0.982					
Married/Stable union	373 (54.8)	62 (16.6)	311 (83.4)		1			
Single/Separated/Widowed	308 (45.2)	51 (16.6)	257 (83.4)		0.99 (0.66−1.49)			
Use of illicit drugs in life/(NI = 6) ***				0.518				
No	658 (97.3)	111 (16.9)	547 (83.1)		1			
Yes	18 (2.7)	2 (11.1)	16 (88.9)		0.62 (0.14−2.72)			
Use of alcohol in the last 12 months/(NI = 6) ***					0.225			
No	492 (72.8)	77 (15.7)	415 (84.3)		1			
Yes	184 (27.2)	36 (19.6)	148 (80.4)		1.31 (0.85−2.03)			
History of incarceration/(NI = 7) ***					0.141			
No	635 (94.1)	102 (16.1)	533 (83.9)		1			
Yes	40 (5.9)	10 (25.0)	30 (75.0)		1.74 (0.83−3.67)			
Age of first sexual intercourse/(NI = 17) ***					0.087			
>=16 years	491 (73.8)	74 (15.1)	417 (84.9)		1			
<15 years	174 (26.2)	36 (20.7)	138 (79.3)		1.47 (0.94−2.29)			
Number of sexual partners in the last 12 months/(NI = 8) ***				0.627				
Did not have sexual intercourse	244 (36.2)	36 (14.8)	208 (85.2)		1			
One partner	361 (53.6)	62 (17.2)	299 (82.8)		1.20 (0.77−1.87)			
Two or more partners	69 (10.2)	13 (18.8)	56 (81.2)		1.34 (0.66–2.70)			
Condom use at last sexual intercourse/(NI = 25) ***				0.571				
Yes	71 (10.8)	10 (14.1)	61 (85.9)		1			
No	586 (89.2)	98 (16.7)	488 (83.3)		1.23 (0.61−2.47)			
Use of lubricant/NI (n = 78) ***			0.235					
Yes	154 (25.5)	21 (13.6)	133 (86.4)		1			
No	450 (74.5)	80 (17.8)	370 (82.2)		1.37 (0.81−2.30)			
Received money or paid in exchange for sex/(NI = 9) ***				0.046				
No	585 (86.9)	90 (15.4)	495 (84.6)		1			
Yes	88 (13.1)	21 (23.9)	67 (76.1)		1.72 (1.01−2.96)			
Sexual intercourse with a person of the same sex/(NI = 11) ***				0.017			0.059	
No	659 (98.2)	105 (15.9)	554 (84.1)		1			1
Yes	12 (1.8)	5 (41.7)	7 (58.3)		3.77 (1.17−12.10)			3.15 (0.96−10.35)
History of sexual intercourse with a sex worker/(NI = 10) ***				0.006			0.013	
No	572 (85.1)	85 (14.9)	487 (85.1)		1			1
Yes	100 (14.9)	26 (26.0)	74 (74.0)		2.01 (1.22−3.33)			1.94 (1.15−3.28)
History of genital sores/(NI = 5) ***				0.038				
No	611 (90.3)	96 (15.7)	515 (84.3)		1			
Yes	66 (9.7)	17 (25.8)	49 (74.2)		1.86 (1.03−3.37)			
History of blisters on genitalia/(NI = 5) ***				0.169				
No	641 (94.7)	104 (16.2)	537 (83.8)		1			
Yes	36 (5.3)	9 (25.0)	27 (75.0)		1.72 (0.79−3.77)			
History of genital warts/(NI = 6) ***				0.103				
No	628 (92.9)	100 (15.9)	528 (84.1)		1			
Yes	48 (7.1)	12 (25.0)	36 (75.0)		1.76 (0.89−3.50)			
Victim of sexual violence/(NI = 4) ***				0.311				
No	644 (94.8)	105 (16.3)	539 (83.7)		1			
Yes	35 (5.2)	8 (22.9)	27 (77.1)		1.52 (0.67−3.44)			
Hospitalization history/(NI = 2) ***				0.777				
No	62 (9.1)	11 (17.7)	51 (82.3)		1			
Yes	618 (90.9)	101 (16.3)	517 (83.7)		0.91 (0.46−1.80)			
Seen a dentist/(NI = 11) ***				0.851				
No	344 (51.3)	56 (16.3)	288 (83.7)		1			
Yes	327 (48.7)	55 (16.8)	272 (83.2)		1.03 (0.69−1.56)			
History of hemodialysis/(NI = 7) ***				0.650				
No	671 (99.4)	111 (16.5)	560 (83.5)		1			
Yes	4 (0.6)	1 (25.0)	3 (75.0)		1.68 (0.17−16.32)			
History of blood transfusion/(NI = 7) ***				0.176				
No	560 (83.0)	88 (15.7)	472 (84.3)		1			
Yes	115 (17.0)	24 (20.9)	91 (79.1)		1.41 (0.85−2.34)			
Shared sharp personal hygiene object/(NI = 2) ***				0.671				
No	316 (46.5)	50 (15.8)	266 (84.2)		1			
Yes	364 (53.5)	62 (17.9)	302 (82.1)		1.09 (0.73−1.64)			

* OR—odds ratio; ** CI95%—confidence interval of 95%; *** NI—no information; **** Adjusted for “city of origin,” **** Hosmer–Lemershow test (*p*= 0.9049) and ROC curve with area under the curve of 0.5578.

**Table 2 vaccines-11-00597-t002:** Characteristics of the 147 participants who reported three doses of the hepatitis B vaccine, according to the vaccine response, Goiás, 2017–2019.

Variables	Isolated anti-HBs Titers			*p*
	Total n = 147 (%)	<10 mUI/mL n = 101 (%)	≥10 mUI/mL n = 46 (%)	
Sex				0.092
Male	46 (31.3)	36 (78.3)	10 (21.7)	
Female	101 (68.7)	65 (64.4)	36 (35.6)	
Age				0.002
50–59 years old	59 (40.1)	32 (54.2)	27 (45.8)	
≥60 years	88 (59.9)	69 (78.4)	19 (21.6)	
Self-declared color				0.067
White	61 (41.5)	37 (60.7)	24 (39.3)	
Brown and Black	85 (57.8)	63 (74.1)	22 (25.9)	
Yellow (Eastern) and Red (Indigenous)	1 (0.7)	1 (100.0)	0	
Body Mass Index (kg/m2) (NI = 43) *				0.464
Malnutrition (<22)	14 (13.4)	9 (64.3)	5 (35.7)	
Eutrophy (22–26)	48 (46.2)	30 (62.5)	18 (37.5)	
Obesity (>27)	42 (40.4)	30 (71.4)	12 (28.6)	
Systemic Arterial Hypertension (NI = 1) *				0.007
No/Don’t know	62 (42.5)	35 (56.5)	27 (43.5)	
Yes	84 (57.5)	65 (77.4)	19 (22.6)	
Diabetes Mellitus				0.760
No/Don’t know	116 (78.9)	79 (68.1)	37 (31.9)	
Yes	31 (21.1)	22 (71.0)	9 (29.0)	
Never smoked a cigarette/tobacco (NI = 2)				0.072
No	111 (76.6)	74 (66.7)	37 (33.3)	
Yes	34 (23.4)	25 (73.5)	9 (26.5)	
Currently smokes cigarettes/tobacco (NI = 2)				0.452
No	111 (76.6)	74 (66.7)	37 (33.3)	
Yes	34 (23.4)	25 (73.5)	9 (26.5)	
Time after last vaccine dose (years)				0.783
	147	101 (68.7)	46 (31.3)	
Mean (standard deviation)		5.93 (5.003)	5.74 (4.915)	

* NI—no information; two participants were excluded from the analysis because they had the date of the last vaccine dose < 60 days from the date of collection.

**Table 3 vaccines-11-00597-t003:** Sociodemographic characteristics of participants aged ≥ 50 years who received the Intervention Regimen or the Comparison Regimen of hepatitis B vaccine, Goiás, 2017–2020.

Variables	Total n = 434 (%)	Intervention Regimen (40 mg/mL) n = 217 (%)	Comparison Regimen (20 mg/mL) n = 217 (%)	*p*
Sex				0.466
Male	133 (30.6)	63 (47.4)	70 (52.6)	
Female	301 (69.4)	154 (51.2)	147 (48.8)	
Age 63.9 (8.8) *				0.920
50–59 years old	157 (36.2)	78 (49.7)	79 (50.3)	
60–97 years old	277 (63.8)	139 (50.2)	138 (49.8)	
Marital status/(NI = 1) **				0.965
Married/Stable Union	239 (55.2)	119 (49.8)	120 (50.2)	
Single/Separated/Widowed	194 (44.8)	97 (50.0)	97 (50.0)	
Education 6.1 (4.6) */(NI= 5) **				0.607
<5 years	207 (48.3)	102 (50.7)	102 (49.3)	
5–9 years	116 (27.0)	56 (48.3)	60 (51.7)	
>9 years	106 (24.7)	56 (52.8)	50 (47.2)	
Self-declared race				0.181
White	139 (32.0)	76 (54.7)	63 (45.3)	
Brown and black	286 (65.9)	137 (47.9)	149 (52.1)	
Yellow (Eastern) and Red (Indigenous)	9 (2.1)	4 (44.4)	5 (55.6)	
Systemic Arterial Hypertension/(NI = 2) **				0.568
No	163 (37.7)	84 (51.5)	79 (48.5)	
Yes	269 (62.3)	131 (48.7)	138 (51.3)	
Diabetes Mellitus/(NI = 3) **				0.375
No	331 (76.8)	169 (51.1)	162 (48.9)	
Yes	100 (23.2)	46 (46.0)	54 (54.0)	
Body Mass Index (kg/m2)/(NI = 17) *				0.854
Malnutrition (< 22)	69 (16.6)	35 (50.7)	34 (49.3)	
Eutrophy (22–27)	166 (39.8)	83 (50.0)	83 (50.0)	
Obesity (>27)	182 (43.6)	90 (50.5)	92 (49.5)	
Ever smoked cigarette/tobacco in life/(NI = 43) *				0.277
No	219 (56.0)	105 (47.9)	114 (52.1)	
Yes	172 (44.0)	92 (53.5)	80 (46.5)	
Current smoker/tobacco/(NI = 11) *				0.683
No	366 (86.5)	182 (49.7)	184 (50.3)	
Yes	57 (13.5)	30 (52.6)	27 (47.4)	

* mean (standard deviation); ** NI—no information.

**Table 4 vaccines-11-00597-t004:** Immunogenic response to hepatitis B vaccine according to Intervention Regimen or Comparison Regimen, Goiás, 2017–2020.

Parameter	Intervention Regimen (40 mg/mL) n = 124		Comparison Regimen(20 mg/mL)n = 114		*p*
	n (%)	IC 95% *	n (%)	IC 95% *	
Seroconversion					0.952
0 mUl/mL	1 (0.8)	(0.0−3.0)	1 (0.9)	(0.2−4.1)	
≥1 mUl/mL	123 (99.2)	(95.6−99.9)	113 (99.1)	(95.2−99.8)	
Seroprotection					0.007
<10 mUl/mL	5 (4.0)	(1.7−9.1)	16 (14.0)	(8.8−21.6)	
≥10 mUl/mL	119 (96.0)	(90.9−98.3)	98 (86.0)	(78.4−91.2)	
GMT **					0.007
	518.2	(456.0–580.4)	260.2	(183.5−336.9)	

* CI95%—confidence interval of 95%; ** Geometric Mean Titer.

## Data Availability

Not applicable.

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
