# Peer review of "Prevalence, Risk Factors and Vaccine Response against Hepatitis B in People Aged 50 Years or Older"

_vaccines, 2023, doi:10.3390/vaccines11030597_

Round 1

Reviewer 1 Report

In general, the manuscript is informative, easy to read and understand. However, there are two issues (Study Sample, Ethics) which should be resolved before further consideration:  

Line 66: Add a new paragraph in which the rationale for conventional dose and boosted dose will be clarified, with appropriate references.  

Lines 72-86: Was written consent provided? Was the consent to participate in the study voluntary? Did the participants receive monetary or any other compensation for participating in this study? How was the sample selected? Specify what type of sample? How was the recruitment of participants carried out? What is the participation rate? What is the response rate? What were the reasons for refusing to participate in the study?  

Line 101: Indicate the place/institution where the interview was conducted?  

Line 105: Indicate the place/institution where the material was taken and where the laboratory test was carried out.  

Lines 171-173: Where in this manuscript are the results presented in the manner stated in this paragraph?  

Line 200: Correct the title of Table `, in accordance with the displayed results (Was only bivariate analysis or bivariate analysis / multiple analysis displayed?).  

Page 7/8: Check whether the variable `History of genital sores / (NI=5)***' is listed twice on Table 1, with different data.  

Lines 237-246: Display these results on a Table.  

Lines 367-368: Informed consent is not sufficient for this study (`Randomized and controlled clinical trial`). Was it written consent? Was it written voluntary consent? Indicate and describe in Methods?       

Reviewer 2 Report

The study has been done cleanly, however the bigger sample size would have helped in drawing better conclusions.

The authors should discuss and mention if any side effects were seen during the Intervention study.

Reviewer 3 Report

Estimated Authors,

I've read with great interest your research paper. In this study, da Cuhna Rosa et al. have reported on the prevalence of anti-HBs antibodies among residents from the region of Goiandira (Brazil). In their analyses, several risk factors were identified, most of them associated with sexual habits of the respondents. Moreover, the present study also reports on the role of a reinforced booster dose (40 µg vs. 20 µg), and its efficacy was appropriately documented.

The study is clearly of substantial interest as, while several countries are facing an even aging general population, interventions able to improve the EFFECTIVE vaccination  rates are welcome. 

From my point of view, the present study could be accepted for publication after some minor improvements:

1) according to the sample size calculation, while the first "half" of the paper is based on a sufficient population, the next one is clearly underpowered; Authors cannot increase the number of participants they were able to recruit (obviously) but they can discuss and acknowledge this issue in the limits of their study;

2) the Results section is too heavily dependent on the tables; while the latter are well designed (but please, merge all the "table 1 - continued; just in case replicate the header), the main text lacks of the appropriate description;

3) discussion is a little bit too generic, and should discuss in more extensive details not only the main results (rows 268 and following ones) but also the negative results, as common risk factors (i.e. being from minorities, education level, history of incarceration, etc) were deprived of substantial effect on the seropositive status. In other words, it would be of some interest if the Authors could provide some glimpses on the PREVENTIVE interventions that were reasonably put in place in the index region well before this study.

4) Please report (if the data is available) the share of participants having any occupational background in the healthcare settings.

Round 2

Reviewer 1 Report

Thank you for the opportunity to re-review the manuscript ID: vaccines-2223213. The authors have addressed all of the issues highlighted in my review, satisfactorily responded to my questions and provided explanations. The authors made the necessary changes to the manuscript. I believe that the changes they have made have significantly improved the manuscript. Thank you to the authors for their responses to my comments.